# NANOLM: AN AFFORDABLE LLM STUDY BENCHMARK VIA ACCURATE LOSS PREDICTION ACROSS SCALES

## ABSTRACT

High computational cost, data collection, and difficulty in distributed training are the three significant barriers in pre-training large language models (LLMs) for many researchers. In this paper, we try to solve the question, "Under constrained computational resources, what type of model design (*e.g.,* model size, model architecture) should I train in order to achieve the best possible performance?" To answer this question, based on Scaling Laws for LLM, we introduce nanoLM: an affordable LLM Study Benchmark via Accurate Loss Prediction across scales. This benchmark unlocks a new LLM study paradigm without direct training. Under the loss basin area, the training loss and model size can be accurately fitted as a power law. This allows us to extrapolate LM from small- to large-scale. For example, with just $13.1\%, 14.2\%$ of the total pretraining cost, we can accurately forecast the loss for models sized **26B** and **52B**. To ensure compatibility with mainstream Transformer architectures, nanoLM offers support for decoder-only structures (*e.g.,* GPT), encoder-only structures (*e.g.,* BERT), and encoder-decoder structures (*e.g.,* T5). Considering that excessive model parameters might lead to GPU memory overflow, nanoLM also supports data parallelism strategies. Our goal with nanoLM is to empower researchers to make cheap and meaningful comparisons of varying model designs at large scales. We also aspire for our benchmark to serve as a bridge between the academic community and the industry.

## 1 INTRODUCTION

Large Language Models (LLMs) pre-trained on internet-scale data have demonstrated impressive performance on various downstream tasks using a variety of evaluation protocols such as zero-shot, few-shot, and fine-tuning. Modern LLMs are based on the Transformer architecture Vaswani et al. (2017), and can be trained with different unsupervised training objectives. For instance, Decoder-only LLMs are pre-trained with a full language modeling objective, computing loss on all tokens, while encoder-decoder models use a masked language modeling objective Wang et al. (2022). Regardless of the pre-training objective used, researchers typically rely on the validation pre-training loss as an evaluation criterion when pre-training language models. This is because there's a signif-

Table 1: Current LLMs. We show five of the current popular transformer models, their model sizes, and the number of training tokens. GPT3 and MT-NLG have a model size exceeding one hundred billion, with training data around 300B. In contrast, other models are smaller in size but have been trained on data surpassing a trillion.

| Model | Params | Tokens |
|-------|--------|--------|
| GPT-3 | 175 B | 300 B |
| MT-NLG | 530 B | 270 B |
| Chinchilla | 70 B | 1.4 T |
| LLAMA 2 | 7/13/65 B | 2 T |
| Falcon | 7/40 B | 1.5/1 T |

icant correlation between pre-training loss and the performance on downstream tasks, even though the latter is challenging to assess comprehensively Brown et al. (2020). OpenAI proposed scaling laws Kaplan et al. (2020) to guide the training of LLM. That is compute-efficient training that involves training very large models on a relatively modest amount of data. As a result, the field has been training larger and larger models, expecting performance improvements, as detailed in Table

1, GPT-3 Brown et al. (2020) and Megatron-Turing NLG (MT-NLG) Smith et al. (2022) show this trend. While DeepMind revisits the trade-off between model size and training tokens by training Chinchilla Hoffmann et al. (2022). Compared to an LLM with the same budget, Chinchilla has a smaller model, more data, and better performance. Chinchilla Hoffmann et al. (2022) argue that the optimal behavior might be scaling data more than scaling model size. The subsequent LLM, LLAMA Touvron et al. (2023b) and Falcon Almazrouei et al. (2023), allocated more computational power to data processing, see Table 1. Despite these discoveries, for average researchers, preprocessing large-scale data and comparing and selecting among various model structures and algorithms at a grand parameter scale remains a costly endeavor. To allow researchers to effectively compare different LLM structures and algorithms within limited computational resources, we need to establish a cost-effective LLM study benchmark. That is the goal of this paper.

**Our contributions.** Our main contribution is to introduce the nanoLM, an affordable LLM study benchmark via accurate loss prediction across scales. We substantiate this contribution as follows:

- We unlock a new LLM study paradigm without direct training. This method can directly compare different model structures and non-$\mu$Transferable parameters on large scales via loss prediction. To ensure compatibility with mainstream Transformer architectures, nanoLM offers support for decoder-only structures (*e.g.,* GPT, LLAMA), encoder-only structures (*e.g.,* BERT), and encoder-decoder structures (*e.g.,* T5). Considering that excessive model parameters might lead to GPU memory overflow, nanoLM also supports data parallelism strategies. (Section 3.1)

- We publicly release a 100B, 400B, 1T, 2T tokens training dataset, chosen from existing LLMs and sorted into various specialized domains. (Section 3.2)

- We successfully utilized our method to forecast the loss for 12-layer GPT, LLAMA, BERT, and T5 models on the C4 and MC4 dataset, near the loss basin. To evaluate our capability to precisely predict loss on more expansive models and datasets, we scaled the GPT model to 32 and 64 layers, culminating in sizes of 26B and 52B, respectively. Subsequent results indicate that the actual loss remains predictable. (Section 4.2)

- By generating a series of small models, with sizes ranging from 38M to 3.4B, and incurring only $13.1\%, 14.2\%$ of the total pretraining cost, nanoLM can accurately forecast the loss for models sized 26B and 52B. This demonstrates that nanoLM can help researchers to make cheap and meaningful comparisons between varying model architectures and serve as a new benchmark for LLM study. (Section 4.3)

To foster reproducibility, we open-source all our code, and data of nanoLM benchmark.

## 2 CAN WE REDUCE THE COST OF PRE-TRAINING LLMS?

We start with the motivation of reducing the cost of pre-training LLMs. While Transformers family models like BERT, GPT, LLAMA series *etc.*, have produced a significantly better performance with scale, the costs of training large models have become increasingly expensive for both data and computational resources. For instance, Meta released its open-source large language models called LLAMA 2 Touvron et al. (2023b), a collection of pre-trained and fine-tuned generative text models ranging in scale from 7 billion to 70 billion parameters. Time to train the 7B, 13B & 70B variants is reported as 184k, 368k & 1.7M GPU hours with A100-80GB. On the other hand, the training corpus of LLAMA includes a new mix of data from publicly available sources with roughly 2 trillion tokens, which is $> 300x$ for the entire English Wikipedia. What's more, to train the larger models without running out of memory, the researcher uses a mixture of model parallelism within each matrix multiply and model parallelism across the layers of the network. In summary, (a) high computational cost, (b) data collection, and (c) difficulty in distributed training are the three major challenges in pre-training large models for many researchers. Previously, the entry point to participate in NLP research was to have a couple of GPUs for training or finetuning. Now, the entry level is to be able to regularly distributedly train or infer a $7 - 70$ billion params model on a couple hundred billion tokens. To this end, we propose nanoLM to reduce the cost of pretrained large language models. In the remainder of this section, we review the related work of how to reduce the pretraining cost and give a theoretical hypothesis for our benchmark nanoLM.

## 2.1 RELATED WORK

Methods to reduce pre-training costs usually involve hardware innovation and training technique improvement. Perrone et al. (2018); Stoll et al. (2020); Salinas et al. (2020) explored transfer learning of HP tuning. Megatron-LM Shoeybi et al. (2019) investigated parallel architecture and got 25% efficiency on 512 GPUs with 8-way model parallelism when training an 8.3B parameter model. Our work is more close to HP transfer learning. The drawback lies in the fact that the mentioned transfer learning occurs between different tasks or datasets for HP transfer. Instead, our benchmark nanoLM predominantly involves grid searching with smaller models, utilizing the loss values of these smaller models to precisely predict the loss of larger models. Next, we give the theoretical study for our benchmark nanoLM.

## 2.2 ESTIMATING HYPERPARAMETERS FOR LARGE MODELS

The term "scaling laws" in modern deep learning aims to predict relations between the performance of models (usually the test loss or some performance metric for fine-tuning tasks) and properties of the model architecture (like model size, dataset size, or training compute) Hestness et al. (2017); Kaplan et al. (2020); Villalobos (2023); Rosenfeld et al. (2020). These relations give researchers insight into understanding network design at a small scale and for optimal computing budget and trade-off design in general. However, these relations must be established through hyperparameter tuning, such as learning rate adjustments, across all model scales. Often, the optimal hyperparameter choices are not known before training large models begins.

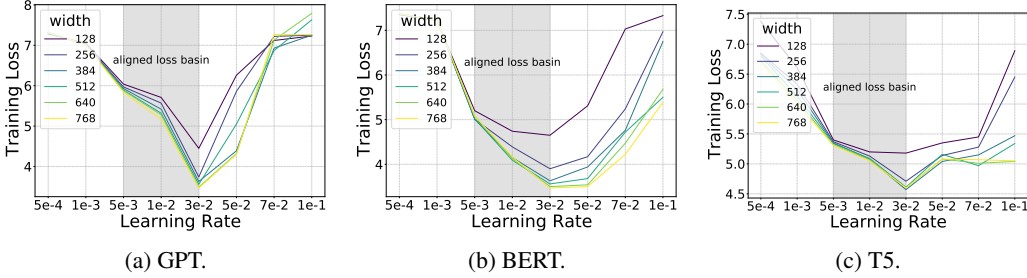

|  |  |  |
|---|---|---|
| (a) GPT. | (b) BERT. | (c) T5. |

Figure 1: Loss during training in relation to the learning rate for different transformer architectures (*i.e.,* GPT, BERT, T5) across widths under $\mu$P. This shows the loss landscapes are aligned for models with different widths.

Tensor Programs, that first introduced in Yang (2019) and later expanded upon in Yang et al. (2022) are developed as a theoretical framework to analyze the infinite-width limits of any architecture. Tensor Programs Yang et al. (2022) proved theoretically that for certain classes of HPs, their optimal values can be well-transferred among all model widths with a unique mapping named Maximal Update Parametrization ($\mu$P), see Figure 1. However, it can not predict the loss values themselves under these HPs but only knows which HPs are better. Later, Yao & Wang (2023) propose a loss prediction method called $\mu$Scaling. They accurately predict loss for arbitrary width by grid-searching for the optimal HP only once on a small width based on $\mu$-transferable HPYang et al. (2022).

Following the above theoretical and empirical study, we release nanoLM, an LLM study benchmark to accurately predict loss from small- to $\infty$ width models. The **primary distinction** between our work and the above work lies in our differing foundational premises. Rather than merely proposing a new theory or methodology, we have introduced a novel LLM study benchmark framework called nanoLM. This framework accommodates the mainstream Transformer architectures, including decoder-only structures (*e.g.,* GPT, LLAMA), encoder-only structures (*e.g.,* BERT), and encoder-decoder structures (*e.g.,* T5), and supports data parallelism strategies. Unlike $\mu$ Scaling uses 8M-676M GPT models to predict the loss for 1.4B models, nanoLM extends this to 77M-3.4B for predicting loss in 52B models. This scale is 184x larger than Scaling, marking a critical step for LLMs. **With the objective of empowering researchers to make meaningful comparisons between varying model architectures and algorithms under constrained computational resources.** This is achieved by predicting the loss of smaller-scale models, thereby obviating the need for full-scale training.

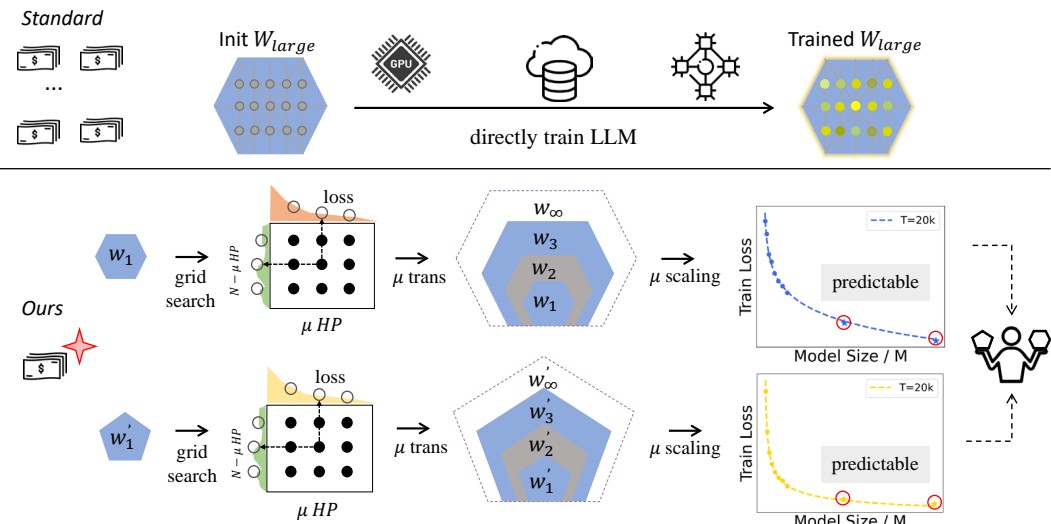

Figure 2: Illustration of standard pretrain LLM way vs. nanoLM. **Top**: Direct pretrain LLM with high computational cost, large data, and distributed training. **Down**: nanoLM accurately predicts the loss across scales. In this scenario, different shapes represent different model architectures. When comparing two LLMs, the process unfolds in four phases: 1) Adjust the width of the two models to narrower dimensions, denoted by $w_1$ and $w'_1$, followed by a grid search on $\mu$ HP. 2) Choose some small widths based on the model architecture, then employ $\mu$P for zero-shot HP Transfer. 3) Train these small-width models, record the losses, and fit the scaling law. 4) With the help of the fitted loss function, directly assess the models at any given width without training the LLM.

## 3 NANOLM

We now introduce the general benchmark that we propose for LLM study without full-scale training called nanoLM. Figure 2 provides the overview of our benchmark. Section 3.1 introduces the methodology in detail, and the performance is discussed in Section 4.

### 3.1 UNLOCK A NEW LLM STUDY PARADIGM WITHOUT DIRECT TRAIN

**Utilizing Pre-training Loss as an Indicator of Model Performance.** LLMs' training involves language modeling tasks, like predicting subsequent or masked tokens. This loss strongly correlates with perplexity. LLaMA Touvron et al. (2023b) also suggests a link between pre-training loss and downstream performance, although this is challenging to assess comprehensively. Thus, pre-training loss generally aligns closely with downstream performance.

$\mu$**P and $\mu$Scaling.** Maximal Update Parametrization *a.k.a.* $\mu$P is a zero-shot transferring function for certain classes of HPs (namely $\mu$Transferable HPs, including learning rate, initialization variance, and multipliers) with model widths changing. For Transformers Vaswani et al. (2017), width is defined as the dimension of embeddings and the inputs/outputs of each block. Tensor Programs theoretically suggest that for $\mu$Transferable HPs, the optimal value $H$ for a small width $w$ and $H'$ for a large width $w'$ satisfy $H' = \mu P(H, r)$, where $r = w'/w$. We illustrate the $\mu$P function we use in Table 2, which corresponds to Yang et al. (2022) Table 8. Please refer to the original papers for theoretical derivations. $\mu$Scaling Yao & Wang (2023) propose a loss prediction method that leverages the observation that $\mu$P and scaling laws are complementary for loss prediction. Based on $\mu$P, $\mu$Scaling fits a power-law function of $L$ with respect to the number of parameters, and expect that this function accurately predicts $L' = L(n)$ that could be achieved by training $M' = M(w_n)$ with HP $H' = H(n)$.

**Extrapolate LM from small- to large-scale.** $\mu$p and $\mu$Scaling are actually "tunnels" towards directly comparing different model structures and non-$\mu$Transferable parameters on large scales via

Table 2: $\mu$P function for a model $M'$ that is r times the widths of M. If a parameter tensor has 2 dimensions that goes infinite when the model width goes infinite, it is "matrix-like" (*e.g.,* a fully-connected hidden layer); if the number is 1 or 0, it belongs to the "others" class. Note that embedding layers are "others". "Output" means the layer that maps an infinite dimension to a finite dimension, which is the word decoding layer ($lm\_head$) in Transformers. A multiplier is a constant multiplied by a parameter tensor, which has a similar function to softmax temperature.

| Hyperparameter (weight) | $M$ | $M' \sim r$ |
|---|---|---|
| AdamW learning rate (matrix-like) | $l$ | $l/r$ |
| AdamW learning rate (others) | $l$ | $l$ |
| Initialization variance (matrix-like) | $\sigma$ | $\sigma/r$ |
| Initialization variance (others) | $\sigma$ | $\sigma$ |
| Multiplier (output) | $\tau$ | $\tau/r$ |
| Multiplier (others) | $\tau$ | $\tau$ |

---

**Algorithm 1** Extrapolate LM from small- to large-scale

---

**Input**: small LM series (different in design) $M = \{M_1, ..., M_n\}$
**Output**: target LLM with lowest loss

  1: For each model design $M_i \in M$, search for the optimal $\mu$ HP on small width.
  2: Generate some models varying small widths only and compute their $\mu$ HPs using $\mu$ transfer.
  3: Train above small-width models, fit power laws, and predict losses for large-width scenarios.
  4: Repeat 1 to 3, select the model with the lowest loss under a large width.

---

loss prediction. We formalize the new LLM study paradigm in Algorithm 1, which can extrapolate LM from small- to large scale without direct training.

**Support Architectures.**   The proposed nanoLM supports three popular transformer architectures: decoder-only structures (*e.g.,* GPT Brown et al. (2020), LLAMA Touvron et al. (2023a)), encoder-only structures (*e.g.,* BERT Devlin et al. (2019)), and encoder-decoder structures (*e.g.,* T5 Raffel et al. (2020)). Each leaves an indelible mark in the field of natural language processing. Furthermore, considering the potential GPU memory overflow due to excessive model parameters, nanoLM integrates with Fully Sharded Data Parallel Zhao et al. (2023). This is a type of data parallel paradigm that enables fitting more data and larger models by sharding the optimizer states, gradients, and parameters.

### 3.2   A PRETRAIN DATA BENCHMARK

To facilitate researchers in using nanoLM for comparative analysis across diverse model designs and improve general cross-domain knowledge of the model, we have carefully curated a selection of pre-training datasets from existing large-scale models (*i.e.,* LLAMA, Falcon, GPT3) and categorized them into various specialized domains. These diverse data sources improve general cross-domain knowledge of the model, as well as downstream generalization capability. The data benchmark dataset we collected encompasses English data with 100B, 400B, 1T, and 2T tokens.

**Data Statistics.**   Our pre-training dataset, as detailed in Appendix Table 5, comprises a rich blend of various open-source materials that span a wide array of domains. Largely, we repurpose data sources previously utilized for training other Large Language Models, adhering strictly to the criteria that the data must be publicly accessible and conducive to open sourcing. The pretraining data for our 100B, 400B, 1T, and 2T versions will all be scaled according to the sampling proportion in Table 5.

**Data Types.**   We added a lot of different domain data to increase the diversity. We have categorized the collected datasets into the following types. *WebText:* The Falcon RefinedWeb Penedo et al. (2023) is a substantial English web dataset derived from CommonCrawl, featuring rigorous filtering and extensive deduplication. OpenWebText2 Gao et al. (2020) is a sizeable dataset of filtered text documents, collected from URLs in Reddit submissions. *Professional Knowledge:* We include

books3 Gao et al. (2020) in the training dataset, which consists of a mix of fiction and nonfiction books. *World Knowledge:* We add English Wikipedia to our training dataset, which is a standard source of high-quality text for language modeling. *Code:* We include the public GitHub dataset and hope to improve downstream performance on code-related tasks. *Academic:* We include arXiv for the training dataset, which consists of preprint research papers Lewkowycz et al. (2022). These papers are mainly in the fields of Math, Computer Science, and Physics. We hope to add scientific knowledge to the dataset. *Question Answering:* We include a dump of Stack Exchange, a website of high-quality questions and answers that covers a diverse set of domains, ranging from computer science to chemistry. We hope that it will improve the question-answering capabilities of downstream models on diverse domains

## 4 EXPERIMENTS

First, we outline our experimental setup in Section 4.1. Then, we present empirical loss prediction findings in Section 4.2. Last, we delve into cost analysis in Section 4.3. For additional analysis comparing errors and insights into the scaling law, please refer to Appendix A.

### 4.1 SETUP

**Model and training details.** To validate the loss prediction capability offered by nanoLM and the data availability of its provided pre-training dataset, we conducted experiments across three dimensions.

Firstly, we conducted a loss prediction on transformer architectures (*i.e.,* GPT, BERT, T5) with 12 layers using the C4 Raffel et al. (2020) dataset on a single machine, single-GPU setup. We utilize a base width of 256 to conduct a grid search for the $\mu$Transferable HP, primarily targeting the learning rate, which has a value range from $5e-4$ to $1e-1$*. The number of parameters ranging from 8M to 700M, see Table 3a.[†] The batch size was established at 16, and the sequence length was set to 2048. According to Yang et al. (2022), when the training steps surpass 5k, the optimal HPs tend to stabilize. To this end, we exit training and fit the loss at 7k steps to save computations as well as study its influences on the predicative performances.

Secondly, to make efficient use of the pretraining data provided by nanoLM (see Section 3.2), nanoLM supports a data parallelism strategy. Specifically, we employed Fully Sharded Data Parallel(FSDP) Zhao et al. (2023) to effectively address the training challenges of larger models. With FSDP, nanoLM is able to efficiently train larger models using fewer GPUs. We conduct loss prediction on 32-layer GPT using pre-training data in Section 3.2. We utilize a base width of 256 to conduct a grid search for the learning rate, ranging from $5e-4$ to $1e-1$. The number of parameters ranges from 38M to 26B; see Table 3b. The batch size was established at 512, and the sequence length was set to 512.

Lastly, to validate the feasibility of $\mu$P and $\mu$Scaling, demonstrating our ability to accurately predict the loss for models with extremely large widths. The experiments for the 64-layer GPT were conducted on Megatron Narayanan et al. (2021). It provides efficient tensor, pipeline, and sequence-based model parallelism for pre-training LLM. We utilize a base width of 256 to conduct a grid search for the learning rate, ranging from $1e-4$ to $1e-2$. The number of parameters ranges from 77M to 52B; see Table 3b. The batch size was established at 512, and the sequence length was set to 2048. We exit training and fit the loss at 10k steps, having processed a total of 10.49B tokens (10% of full pretrain data small version in Section 3.2).

The batches are fed to the model in the same order for all runs. All of our experiments are run on A100 GPUs. More hyperparameters can be found in the Appendix.

---

*The $\mu$HP include learning rate, initialization standard deviation, input&output multipliers. To conserve computational resources, we focused our search solely on the learning rate. However, researchers can still explore the other $\mu$HP on nanoLM to achieve even better results.

[†]Given that the experiments for this dimension were conducted on a single machine with a single GPU, both BERT and T5 ran out of memory when expanded to a width of 2048. Therefore, in this dimension, neither BERT nor T5 has experiments at a width of 2048.

Table 3: Model Parameters and Loss Across Various Widths and Architectures. Note: Loss values in bold represent predicted values.

(a) 12-Layer Model with Parameter Count and Loss value (Parameters in Millions)

| Width | 128 | 256 | 384 | 512 | 640 | 768 | 896 | 1024 |
|---|---|---|---|---|---|---|---|---|
| GPT | 8.82 | 22.36 | 40.61 | 63.59 | 91.28 | 123.69 | 160.82 | 202.67 |
| Loss@20k | 4.45 | 4.2 | 4.05 | 3.94 | 3.90 | 3.87 | 3.85 | 3.84(**3.810**) |
| LLAMA | 9.59 | 25.47 | 47.64 | 76.10 | 110.85 | 151.90 | 199.24 | 252.86 |
| Loss@20k | 4.49 | 4.31 | 4.24 | 4.20 | 4.18 | 4.17 | 4.11 | 4.10(**4.112**) |
| BERT | 22.52 | 51.28 | 86.33 | 127.67 | 175.31 | 229.24 | 289.45 | 355.96 |
| Loss@20k | 3.99 | 3.15 | 2.95 | 2.88 | 2.83 | 2.81 | 2.78 | 2.77(**2.782**) |
| T5 | 26.46 | 67.03 | 121.75 | 190.63 | 273.66 | 370.85 | 482.20 | 607.70 |
| Loss@20k | 4.75 | 4.66 | 4.60 | 4.58 | 4.52 | 4.47 | 4.42 | 4.40(**4.415**) |

(b) 32-Layer and 64-Layer Models: Parameter Count and Loss value (Parameters in Billions).

| Width | 256 | 384 | 512 | 640 | 768 | 896 | 1024 | 2048 | 8192 |
|---|---|---|---|---|---|---|---|---|---|
| 32-layer GPT | 0.038 | 0.076 | 0.126 | 0.189 | 0.265 | 0.353 | 0.454 | 1.714 | 26.185 |
| Loss@7k | 3.92 | 3.76 | 3.65 | 3.59 | 3.54 | 3.49 | 3.47 | 3.45 | 3.41(**3.381**) |
| 64-layer GPT | 0.077 | 0.153 | 0.254 | 0.381 | 0.532 | 0.709 | 0.911 | 3.432 | 52.385 |
| Loss@10k | 3.656 | 3.389 | 3.298 | 3.215 | 3.198 | 3.087 | 3.080 | 2.958 | 2.883(**2.861**) |

$\mu$**P and** $\mu$**Scaling Settings.** All of our models are trained from scratch. Additionally, we adhere to the recommendation by Yang et al. (2022) to initialize the word embeddings and query parameters as all-zero to avoid the Gaussian Process [‡]. We maintain a consistent head dimension of 64 for each attention head and scale the number of attention heads with the model's width. The settings for dropout and weight decay are both configured to 0. Furthermore, we employ the AdamW optimizer Loshchilov & Hutter (2019) with its default configurations.

## 4.2 EMPIRICAL RESULTS: FITS & EXTRAPOLARION OF LOSS FUNCTION FORMS

Our main goal is to justify the third step in Algorithm 1. We use a base width of 256 to grid-search for the optimal $\mu$Transferable HPs. For 12-layer models, we found the optimal (learning rate, initialization standard deviation, input&output multipliers) being $(3e-2, 0.05, 5.0)$. For 32-layer and 64-layer GPT, the optimal $\mu$ HPs are $(3e-2, 0.04, 4.0)$ and $(1e-3, 0.01, 1.0)$. These HPs are $\mu$Transferred to other widths by Table 2. Some specific loss values can be found in Table 3, while the corresponding fitted curves are illustrated in Figure 3. For additional numerical loss values, please refer to Appendix E. Appendix C provides details on the hyperparameter settings, while the grid-search results can be found in Appendix D.

**12-layer models on C4 and MC4 with a single GPU.** For GPT, LLAMA, BERT, T5, data points with widths $\leq 896$ [§] are used to fit the power-law curves and predict the loss for widths $> 896$.

All loss values, as well as the $a, b, c$ coefficients in power law $L = aC^b + c$, and their standard deviations computed with $scipy.optimize.curve\_fit()$ [¶]. For the 12-layer GPT and LLAMA, the power-law equation is $L = 2.07 * C^{-0.52} + 3.85$ and $L = 1.34 * C^{-0.49} + 4.02$. Similarly, for the 12-layer BERT, the result is $L = 61.18 * C^{-1.31} + 3.43$; and for the 12-layer T5, it's $L = 0.25 * C^{-0.47} + 2.82$. From Figure 3, it's evident that based on the loss from the initial

---

[‡]A Gaussian Process with may cause misalignment of landscapes between small and large models Yao & Wang (2023)

[§]Based on our experimental results, there's a positive correlation between the number of data points and the accuracy of the fitted loss curve. We choose seven points due to the limited computation resource.

[¶]https://docs.scipy.org/doc/scipy/reference/generated/scipy.optimize.curve_fit.html

narrower models, we successfully predicted the loss of larger-width models at the 7k step. As theoretically proved by $\mu$P Yang et al. (2022), if we continue adjusting the $\mu$ HP according to model width, the losses of even larger models can be directly predicted. Moreover, under $\mu$P, as the model width expands, the training loss demonstrates a progressively decreasing trend, aligning with Yang et al. (2022) conclusion that "wider is better". In contrast, the yellow points, not conforming to the $\mu$P, underperform the same-width networks in nanoLM and do not adhere to the scaling law. Finally, we validated the capacity of nanoLM to predict the loss over extended training steps (20k) for Transformer-like models such as GPT, LLAMA, BERT, and T5. Additionally, its effectiveness in anticipating loss on diverse multilingual datasets, such as MC4 Raffel et al. (2020) (a multilingual, colossal, and cleaned version of the Common Crawl web corpus), has also been established.

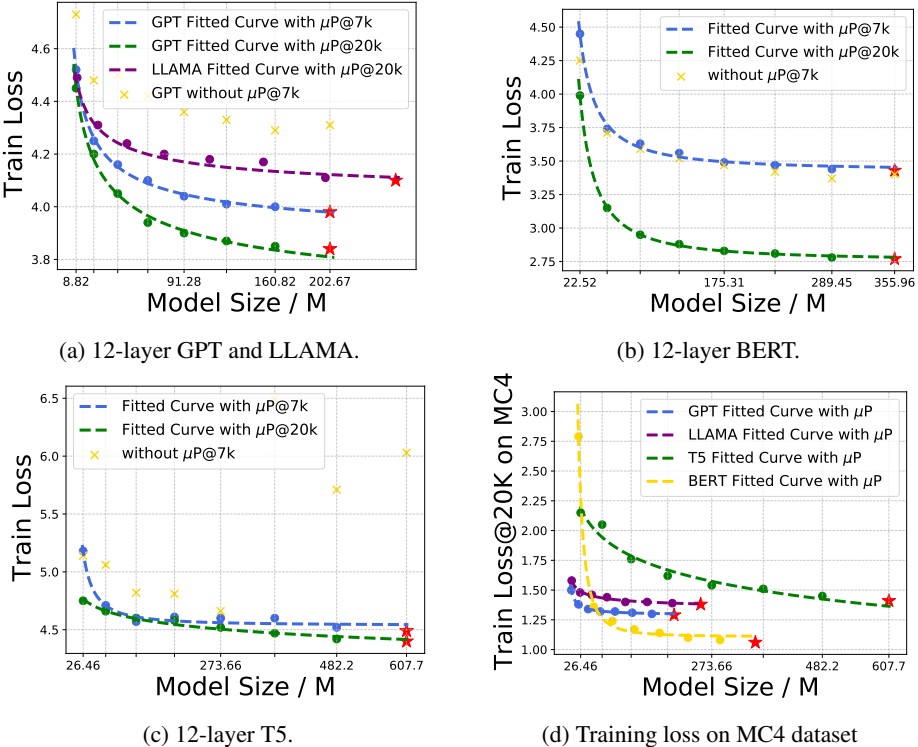

(a) 12-layer GPT and LLAMA.

(b) 12-layer BERT.

(c) 12-layer T5.

(d) Training loss on MC4 dataset

Figure 3: Fitting result with $\mu$P and without $\mu$P: The dots illustrate the training loss across different small widths while incorporating $\mu$P. In contrast, the yellow cross points display the training loss at those very widths but without employing $\mu$P. We fit these dots to adapt the scaling law, aiming to ascertain if the loss of the final one models is consistent with this trend. The red star denotes the actual loss values from our training of the predicted wider models.

**32-layer GPTs on data benchmark with FSDP.** To validate nanoLM loss prediction capability for larger-scale models, we expanded the GPT to 32 layers. On nanoLM enhanced with a data parallel strategy (FSDP), we trained models with widths ranging from 256 to 2048, utilizing training data from Section 3.2. Based on these eight data points, we predicted the loss for a model with a width of 8192. The fitting results are presented in Figure 4a with the power-law equation being $L = 0.077 * C^{-0.61} + 3.37$. Notably, nanoLM successfully predicted the loss of a 26B model at 7k steps based on the losses from models ranging from 38M to 1.7B. This demonstrates nanoLM remarkable ability to accurately predict losses even when scaling up to models of billion-scale magnitude.

**64-layer GPTs on data benchmark with Megatron.** To validate the correctness of $\mu$P Yang et al. (2022) and $\mu$Scaling Yao & Wang (2023), especially in predicting the loss of super-large scale models, we expanded the GPT layers to 64 and trained using Megatron Shoeybi et al. (2019), which supports Data Parallel, Pipeline Parallel, and Tensor Parallel. We trained models with widths ranging from 256 to 2048 using the data from Section 3.2 and used these eight data points to forecast the loss

for a model with a width of 8192. The fitting outcomes are presented in Figure 4b with the power-law equation being $L = 0.249 * C^{-0.47} + 2.82$. The results show that we can still successfully predict the loss of a 52B model at 10k steps based on the losses from models ranging from 77M to 3.4B, further affirming the accuracy of $\mu$P and $\mu$Scaling.

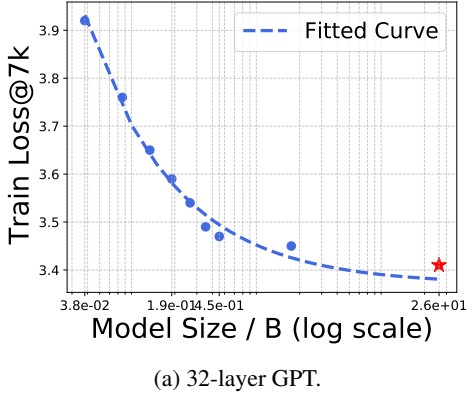

(a) 32-layer GPT.

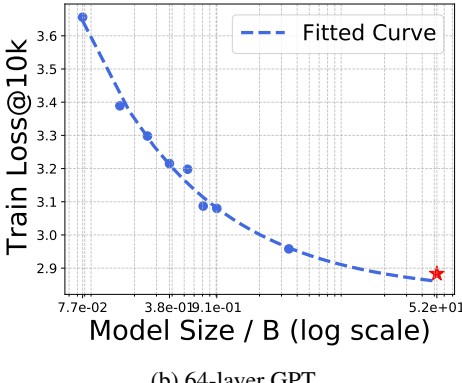

(b) 64-layer GPT.

Figure 4: Fitting results under $\mu$P: The blue dots signify the loss values of the small-width models employed to fit the scaling law. In contrast, the red star denotes the actual loss values from our training of the predicted wider models.

### 4.3 EFFICIENCY

**Tuning Cost Ratio.** We calculate the number of floating point operations (FLOPs) in a model following Narayanan et al. (2021). Given a language model with $l$ transformer layers, width *a.k.a.* hidden size $w$, sequence length $s$, vocabulary size $V$, and training batch size $B$. The total number of floating-point operations is $M(l, w, s, V, B) = 96Bslw^2 \left(1 + \frac{s}{6w} + \frac{V}{16lw}\right)$. Thus, the ratio of tuning cost to pretraining cost in FLOPs from small to large scale can be approximated as

$$\frac{M(l, w_1, s, V, B) * t + \sum_{i=2}^{n} M(l, w_i, s, V, B)}{M(l, w_t, s, V, B)} \tag{1}$$

where $w_1$ is small model width and $t$ denotes corresponding number of grid search trails on $\mu$Transferable HPs (as detailed in Table 2). $n$ signifies the count of distinct width models used for the fitting loss function, while $w_t$ is the target model width. In our experiments, the model architecture comprises a 32-layer GPT with a sequence length of 512 and a batch size of 512. The vocabulary is based on GPT4 and encompasses 100,256 tokens. We generated nine small-width models, ranging from 128 to 1024, to fit the loss of the 8192-width model; the ratio of tuning cost is approximated at 0.131. Using a similar calculation method, the 64-layer GPT tuning cost ratio is 0.142. In simpler terms, with just $13.1\%, 14.2\%$ of the total pretraining cost, we can accurately forecast the loss for models sized 26B and 52B.

## 5 CONCLUSION AND FUTURE WORK

We present nanoLM, a cost-effective benchmark for Large Language Model (LLM) studies, designed to predict losses accurately across various scales without needing direct training. This method allows for comprehensive comparisons of different model architectures and algorithms, supporting GPT, LLAMA, BERT, and T5, and works well with Fully Sharded Data Parallel. Our curated English pre-training datasets, derived from existing LLMs and classified by professional fields, offer token counts ranging from 100B to 2T. nanoLM's effectiveness in predicting losses has been confirmed through three types of experiments: a single-machine, single-GPU setup on the C4 and MC4 dataset; a multi-GPU, multi-machine setup with our dataset benchmark using parallelism strategies; and a large-scale Megatron variant. These experiments demonstrate nanoLM's ability to predict losses across scales, enabling researchers with limited resources to compare different models efficiently. Moving forward, we aim to extend nanoLM to more languages and other task (*e.g.,* vision), fostering LLM advancements, reducing resource waste from non-transferable findings, and enhancing collaboration between academia and industry.

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

# A    MORE ANALYSIS

**Error Comparison of Scaling Laws.**    Based on Table 8 of the Cerebras-GPT Dey et al. (2023) and the scaling law's power function $L = ax^b + c$, we conducted an error comparison analysis of the predicted losses for Cerebras-GPT Dey et al. (2023) , Pythia Biderman et al. (2023), and nanoLM. As shown in table 4, Cerebras-GPT fitted the 13B model's loss using $111M - 6.7B$ models, Pythia for $12B$ with $7M - 6.9B$, and nanoLM for $52B$ from $77M - 3.4B$, with their respective errors being 0.025, 0.019, and 0.022. When fitting losses for models down to 10B, the errors are 0.034, 0.049, and 0.018, respectively. Additionally, we calculated the covariances of the fitted coefficients $\{a, b, c\}$, finding that nanoLM's covariances are significantly lower than those of Cerebras-GPT and Pythia. These experimental results demonstrate the (a) $\mu$P infinite neural network is theoretically correct and (b) scaling laws are empirically reliable on any scale. (c) The loss prediction of nanoLM is more stable and reliable. Loss prediction validation is costly, and we have conducted as many experiments as possible within our computational power limits (nanoLM has reached 52B for this purpose, while Pythia and Cerebras-GPT have only reached 13B). Further experimentation would be prohibitively expensive in terms of computational costs.

Table 4: Comparison of Fitted Results: The "Coeffs & Cov" denote the coefficients {a, b, c} and the covariance of the power-law function $y = aC^b + C$.

| Model | Size & Loss | | | | | | | | | Coeffs & Cov | | |
|---|---|---|---|---|---|---|---|---|---|---|---|---|
| Cerebras-GPT | 0.111 | 0.256 | 0.59 | 1.30 | 2.700 | 6.700 | 13.00 | - | - | 6.76e1 | -8.45e-2 | 7.25e1 |
| | 2.608 | 2.349 | 2.181 | 1.997 | 1.834 | 1.704(0.034) | 1.572(0.025) | - | - | 4.84e1 | 2.10e-2 | 3.44e-1 |
| Pythia | 0.070 | 0.160 | 0.410 | 1.000 | 1.400 | 2.800 | 6.900 | 12.00 | - | 9.67e6 | -0.34 | 1.42 |
| | 2.549 | 2.204 | 1.989 | 1.858 | 1.889 | 1.724 | 1.644(0.049) | 1.601(0.019) | - | 3.89e7 | 8.89e-2 | 1.50e-1 |
| nanoLM | 0.077 | 0.153 | 0.254 | 0.381 | 0.532 | 0.709 | 0.911 | 3.432 | 5.24e1 | 0.25 | -0.47 | 2.82 |
| | 3.656 | 3.389 | 3.298 | 3.215 | 3.198 | 3.087 | 3.080 | 2.958(0.018) | 2.883(0.022) | 7.33e-2 | 8.50e-2 | 7.66e-2 |

**Embedding counts as model size.**    We demonstrate that our scaling law fits worse if embedding weights are not counted in the model sizes, in contrast to Kaplan et al. (2020).This is potentially because $\mu$P concluded that the learning rate of embedding layers should not be scaled down with widths while Kaplan et al. (2020) searched for a unified learning rate for all layers on each model size, making embeddings learned too slow, and matrix-like parameters dominate the training dynamics.

**Scaling law fail outside the loss basins.**    Theoretically, $\mu$P suggests similar train ing dynamics across different widths for arbitrary HP, but we observe that the scaling laws fit well only in loss basins. According to our follow-up experiments, this observation is regardless of data or training steps and still exists when models are more sufficiently trained. Thus, we suggest searching for the best HPs first anyway.

**Smaller models are more vulnerable.**    We grid-search for the best HPs for 6-layer models with batch size 32 and found $(5e - 4, 0.02, 3.5)$ being inside the loss basin. As shown in Figure 5a (red line), nanoLM works perfectly for this single point. We then explored other points around it and found that the scaling laws have larger deviations than 12-layer models. This is potentially because small models are more vulnerable to slight misalignment of loss landscapes across $\mu$-Transfer. However, we easily balance-off this deviation by fitting scaling laws with the average results across all these HPs near the loss basin. This works perfectly as shown in Figure 5b, and can be very practical in applying nanoLM because we observe in Figure 5a that larger widths (*e.g.,* 2048, 3072) have low variance in training loss w.r.t different HPs.

**General conditions for scaling laws.**    Previous scaling laws directly search for HPs on each scale, and the optimal HPs do not satisfy $\mu$P function. This indicates that $\mu$P is a sufficient but not necessary condition for scaling laws, and scaling law itself may represent a higher level of universality.

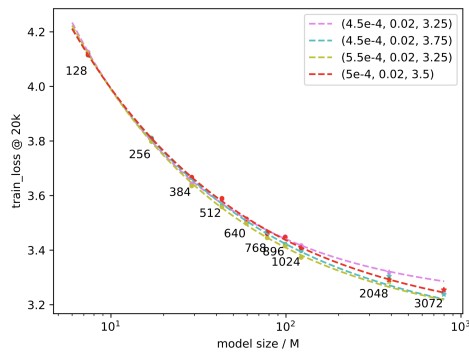
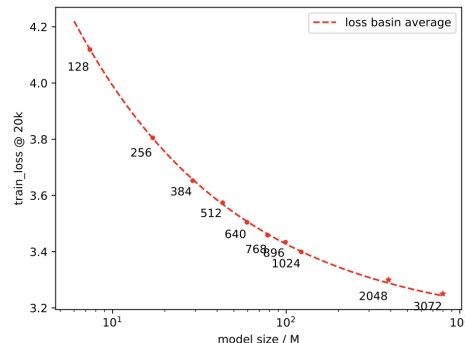

(a) Scaling law for training loss with different HPs for 6-layer models.

(b) Scaling law for average training loss in the loss basin of 6-layer models.

Figure 5: Results with 6-layer Models.

Table 5: Pre-training data ratio.

| Dataset | Sampling prop(%) | Total tokens(B) |
|---|---|---|
| Arkiv | 6.04 | 28.31 |
| Books | 5.22 | 24.46 |
| Falcon RefinedWeb | 20.81 | 97.49 |
| Falcon RefinedWeb(wiki-like) | 49.78 | 233.21 |
| OpenWebText2 | 3.11 | 14.59 |
| StackExchange | 3.81 | 17.84 |
| Github | 10.18 | 47.70 |
| Wikipedia | 1.03 | 4.82 |

## B    PRE-TRAINING DATA RATIO

## C    THE HYPERPARAMETER SETTINGS FOR ALL EXPERIMENTS

The specific parameters of the experiment are as follows. (1) The parameters of the model are: vocab_size = 50304; block_size = 1204; n_layer = [12, 32, 64]; num_heads = 12; dropout = 0.0; output_mult = 1.0; zero_query = True; zero_emb = True. hp_tune_actual_width = [128, 256, 384, 512, 640, 768, 896, 1024, 2048, 4096, 8192]; (2) The parameters of the data are: input_length = 512; mlm_probability = 0.15; mean_noise_span_length = 3.0; num_workers = 2; (3) The parameters of the optimizer are: name = adamwscale; batch_size = [16, 512]; total_steps = [7000, 10000]; warmup_steps = 5000; lr_scheduler = cosine; weight_decay = 0.0; grad_clip = 1.0; grad_acc = 1; final_cosine = 1e-5; base_lr = [5e-4, 1e-3, 5e-3, 1e-2, 3e-2, 5e-2, 7e-2, 1e-1].

## D    BASED ON THE BASIC WIDTH OF 256, THE GRID SEARCH RESULTS

Table 6: grid search on base width = 256. The specific parameters of the experiment are: n_layer = 12, batch_size = 16, hp_tune_actual_width = 256, total_steps = 7000, base_lr = [5e-4, 1e-3, 5e-3, 1e-2, 3e-2, 5e-2, 7e-2, 1e-1].

| lr | 5e-4 | 1e-3 | 5e-3 | 1e-2 | 3e-2 | 5e-2 | 7e-2 | 1e-1 |
|---|---|---|---|---|---|---|---|---|
| 12-layer BERT loss | 7.37 | 7.27 | 5.01 | 4.39 | 3.9 | 4.17 | 5.24 | 6.97 |
| 12-layer GPT loss | 7.3 | 7.03 | 5.97 | 5.57 | 3.74 | 5.86 | 7.22 | 7.25 |
| 12-layer T5 loss | 6.85 | 6.33 | 5.37 | 5.13 | 4.71 | 5.14 | 5.28 | 6.45 |

Table 7: grid search on base width = 256. The specific parameters of the experiment are: n_layer = 64, batch_size = 512, hp_tune_actual_width = 256, total_steps = 10000, base_lr = [1e-4, 5e-4, 7e-4, 1e-3, 3e-3, 5e-3, 7e-3, 1e-2].

| lr | 1e-4 | 5e-4 | 7e-4 | 1e-3 | 3e-3 | 5e-3 | 7e-3 | 1e-2 |
|---|---|---|---|---|---|---|---|---|
| 64-layer GPT loss | 4.35 | 3.73 | 3.69 | 3.64 | 8.37 | 13.3 | 9.66 | 8.12 |

# E SPECIFIC LOSS VALUE

## E.1 NANOLM ON C4

Table 8: training loss on 12-layer@7k steps. The specific parameters of the experiment are: n_layer = 12, batch_size = [16, 512], hp_tune_actual_width = [128, 256, 384, 512, 640, 768, 896, 1024], base_lr = [1e-3, 3e-2].

| width | 128 | 256 | 384 | 512 | 640 | 768 | 896 | 1024 |
|---|---|---|---|---|---|---|---|---|
| BERT w/o $\mu$P | 4.25 | 3.71 | 3.59 | 3.52 | 3.47 | 3.42 | 3.37 | 3.40 |
| BERT with $\mu$P | 4.45 | 3.74 | 3.63 | 3.56 | 3.49 | 3.47 | 3.44 | 3.43 |
| GPT w/o $\mu$P | 4.73 | 4.48 | 4.50 | 4.42 | 4.36 | 4.33 | 4.29 | 4.31 |
| GPT with $\mu$P | 4.52 | 4.25 | 4.16 | 4.10 | 4.04 | 4.01 | 4.00 | 3.98 |
| T5 w/o $\mu$P | 5.14 | 5.06 | 4.82 | 4.81 | 4.66 | 6.50 | 5.71 | 6.03 |
| T5 with $\mu$P | 5.18 | 4.71 | 4.57 | 4.61 | 4.60 | 4.60 | 4.52 | 4.49 |

## E.2 NANOLM ON MC4

Table 9: training loss on 12-layer@20k steps. The specific parameters of the experiment are: n_layer = 12, batch_size = [16, 512], hp_tune_actual_width = [128, 256, 384, 512, 640, 768, 896, 1024], base_lr = [1e-3, 5e-2].

| width | 128 | 256 | 384 | 512 | 640 | 768 | 896 | 1024 |
|---|---|---|---|---|---|---|---|---|
| Bert | 2.79 | 1.36 | 1.24 | 1.17 | 1.14 | 1.10 | 1.08 | 1.06 |
| T5 | 2.15 | 2.05 | 1.76 | 1.62 | 1.54 | 1.51 | 1.45 | 1.41 |
| GPT | 1.50 | 1.38 | 1.34 | 1.32 | 1.32 | 1.31 | 1.30 | 1.29 |
| LLAMA | 1.58 | 1.48 | 1.46 | 1.44 | 1.40 | 1.40 | 1.39 | 1.38 |

## E.3 NANOLM ON PRETRAIN DATA BENCHMARK WITH FSDP

Table 10: training loss on 32-layer@7k steps. The specific parameters of the experiment are: n_layer = 32, batch_size = 512, hp_tune_actual_width = [256, 384, 512, 640, 768, 896, 1024, 2048, 4096, 8192], total_steps = 7000, base_lr = 5e-2.

| width | 256 | 384 | 512 | 640 | 768 | 896 | 1024 | 2048 | 8192 |
|---|---|---|---|---|---|---|---|---|---|
| GPT with $\mu$P | 3.92 | 3.76 | 3.65 | 3.59 | 3.54 | 3.49 | 3.47 | 3.45 | 3.41 |

## E.4 MEGATRON ON PRETRAIN DATA BENCHAMARK

Table 11: training loss on 64-layer@10k steps. The specific parameters of the experiment are: n_layer = 64, batch_size = 512, hp_tune_actual_width = [ 384, 512, 640, 768, 896, 1024, 2048, 8192], total_steps = 10000, base_lr = 1e-3.

| width | 256 | 384 | 512 | 640 | 768 | 896 | 1024 | 2048 | 8192 |
|---|---|---|---|---|---|---|---|---|---|
| GPT with $\mu$P | 3.656 | 3.389 | 3.298 | 3.215 | 3.198 | 3.087 | 3.080 | 2.958 | 2.883 |

