# OpenReview forum: "NanoLM: An Affordable LLM Study Benchmark via Accurate Loss Prediction Across Scales"
_ICLR.cc/2024/Conference — Submitted to ICLR 2024_

### Official Review · Reviewer_Ahkx · 2023-10-27

**Soundness:** 2 fair
**Presentation:** 3 good
**Contribution:** 2 fair
**Rating:** 5
**Confidence:** 3

**Summary:**

The paper introduces nanoLM, a benchmarking framework aimed at making the study of Large Language Models (LLMs) more affordable and accessible. NanoLM leverages the Scaling Laws for LLMs to predict training loss across various model scales, thus allowing for meaningful comparisons between different architectures and configurations without the need for extensive computational resources. The paper validates the utility of nanoLM by accurately forecasting the loss for models with sizes up to 52B while incurring only a fraction of the total pretraining cost, and it supports a range of Transformer architectures and data parallelism strategies.

**Strengths:**

* The authors provide a cost-effective and scalable solution for LLM research by enabling accurate loss prediction across various model scales, thus allowing researchers to bypass the computationally intensive direct training phase. TAuthors tested multiple architectures like GPT, BERT, and T5 models.

* The authors also release large-scale, field-specific datasets for pre-training, with token counts ranging from 100B to 2T. This adds significant value as it enables more nuanced and targeted model comparisons and evaluations.

* The authors conducted extensive validation of nanoLM's capabilities across multiple dimensions, including single-machine, single-GPU setups, as well as multi-GPU, multi-machine configurations.

**Weaknesses:**

* The paper focuses on loss prediction as the primary metric for benchmarking and comparison, but it does not explore how well this loss prediction correlates with performance on various downstream tasks. It's not clear if this is a limitation as loss on a pre-training task is not always indicative of performance in practical applications.

* While nanoLM aims to be a universal benchmark, its current limitation to the English language could restrict its applicability and adoption in global, multi-lingual research communities.

**Questions:**

1. How robust is nanoLM's loss prediction across different types of transformer architectures, beyond GPT, BERT, and T5 structures? Have you tried more recent open-source models like Mistral or Llama? Are there certain architectures or hyperparameter configurations where nanoLM's predictions are less accurate?

2. Have you considered extending nanoLM's capabilities to predict other important metrics beyond loss, such as energy efficiency or training time, to provide a more comprehensive view of a model's trade-offs?

---

> ### Author Response · Authors · 2023-11-17
> **Thanks for your vulable suggestion!**
>
> ### Q1: Pre-training loss correlates with performance on various downstream tasks is not well disscussed in paper.
> Great question! The success of Large Language Models (LLMs) mainly relies on their base models, where the key is the loss value in set datasets. For example, GPT-4 [1] aims for a loss value of 1.25. LLMs are trained on tasks like predicting the next or hidden words, and this loss is closely linked to perplexity. The LLaMA study also hints at a connection between pre-training loss and later performance, but it's hard to fully evaluate. In LLAMA’s Figure 1 [2], you can see training loss decreases in models from 7B to 65B. Figure 2 shows better results in question answering and reasoning with more training tokens. Essentially, lower pre-training loss often means better performance later, highlighting the importance of nanoLM's loss prediction.
> We appreciate your feedback about the unclear analysis. We revised it in Section 3.1 for better clarity.
> ### Q2: How robust is nanoLM's loss prediction ability across different transformer architectures and other language？
> Thank you for your query. Based on the theoretical proofs of $\mu$P [3], nanoLM is well-suited for a range of transformer-like architectures. We've expanded nanoLM's benchmark to include LLAMA and additional languages, demonstrating its robustness in loss prediction. These updates are detailed in the revised Section 4.2 and Figure 3. Looking ahead, we aim to incorporate more popular transformer-like architectures and diverse tasks, such as vision, into nanoLM. We also encourage users to actively engage with nanoLM, contributing to its ongoing development.
> ### Q3: Are there certain architectures or hyperparameter configurations where nanoLM's predictions are less accurate?
> We have confidence in our results for two reasons: (a) $\mu$P is theoretically sound, and (b) scaling laws have shown empirical reliability across various scales. We've included additional analysis to assess nanoLM's loss prediction accuracy:
> (1) Embedding counts in the model size. Our new experiment shows that our scaling law fits less accurately if embedding weights are excluded, unlike in OpenAI established scaling law [4]. This could be due to $\mu$P's finding that embedding layer learning rates shouldn't be scaled down with widths. Our unified learning rate for all layers at each model size might result in slower learning for embeddings, with matrix-like parameters predominantly influencing training dynamics.
> (2) We recommend initializing the word embedding and query parameters to all zeros to avoid Gaussian Processes. Accoding to $\mu$P thereactical proofs, a Gaussian Process with may cause misalignment of landscapes between small and large models.
> (3) Based on our current experiments, nanoLM performs well across different transformer-like architectures (decoder-only, encoder-decoder, encoder). However, we encourage users to test nanoLM on other structures as part of future work.
> We appreciate your suggestion and these analyses is added in a new appendix A of the paper.
> ### Q4: Predict other metrics beyond loss (e.g., energy efficiency or training time).
> This is a good catch! As noted in Chichilla's[5] paper, both training time and energy efficiency can be expressed as functions of model size, exhibiting a linear relationship. We will add this clarification in the new appendix, and apologize for any confusion caused. Thank you for highlighting this important aspect.
> ### ref
> [1] https://arxiv.org/pdf/2303.08774.pdf
> [2] https://arxiv.org/pdf/2302.13971.pdf
> [3] https://arxiv.org/abs/2011.14522
> [4] https://arxiv.org/abs/2001.08361
> [5] https://arxiv.org/pdf/2203.15556.pdf

---

> ### Author Response · Authors · 2023-11-22
> **Inquiry on Further Questions for nanoLM Review Before Discussion Deadline**
>
> Dear Reviewer Ahkx,
>
> Thank you for your review comments. Do you have any further questions or concerns regarding nanoLM? As we are approaching the deadline for the author-reviewer discussion period, we would like to address any remaining queries before this date.
>
>
> Best regards,
> The authors

---

### Official Review · Reviewer_Fxnv · 2023-10-29

**Soundness:** 3 good
**Presentation:** 2 fair
**Contribution:** 3 good
**Rating:** 6
**Confidence:** 4

**Summary:**

This paper builds up a benchmark for evaluating LLM’s performance without direct training. The mainstream transformer architectures and data parallelism are supported. Empirical results demonstrate that the proposed benchmark can accurately predict the loss of models after scaling up.

**Strengths:**

- This paper proposes a cost-efficient benchmark to study LLM’s performance
- The authors organize the publicly accessible datasets for training LLMs

**Weaknesses:**

- To make the paper stronger, the authors should phrase the critical component of the method more clearly.
- In terms of method, the paper lacks its own insights into the scaling law. The two main components uP and uScaling are refer to others’ work and not discussed enough

**Questions:**

Algorithm 1 should be worded more clearly:
- Does “different in design” mean each input model is different from not only the widths but also the architecture?
- Line 2: “Generate some models varying widths only” reads confusing. Does it mean generating models for each M_i by varying the widths? Is there any limit to varying the width?
- Line 3: “Train above small-width models” reads as the width varied in line 2 should be small. So stating it clearly in Line 2 would help readers to understand the whole algorithm.

Other questions:
- In Figure 4, the training loss at 7k and 10k of different sizes of models are shown. 7k and 10k iterations are very early stages of 26B and 52B models. What about the loss prediction accuracy for longer training steps(eg, 20k or more)?

- The author should put the loss prediction values(with multiple fitting and std included) and ground truth into a table and display in the main paper. So the readers have a clear sense of the robustness and performance of the proposed benchmark.

Minor: there are some inconsistent notations and grammar errors, please fix them accordingly.

---

> ### Author Response · Authors · 2023-11-17
> **Thank you for the positive recommendations!**
>
> Thank you for the positive recommendations and valuable feedback!
> ### Q1: Ambiguity in Algorithm 1.
> * Yes, line 1 "different model design" includes width and architecture variations, as shown in our paper's Figure 2. For example, $M_1$ uses Llama, and $M_2$ GPT. nanoLM allows for loss predictions and comparisons in these models at larger sizes without full-scale training.
> *  Yes, line 2 "Generate some models varying widths only" mean generating models for each $M_i$ by varying the widths and there is no limit to varying the width according $\mu$P [1] condition.
> *  Thanks for your suggestion, we implemented it for the revised version of Alg 1 Line 2.
> ### Q2: What about the loss prediction accuracy for longer training steps(eg, 20k or more)?
> Thank you for your valuable suggestion. We've now increased the training of our 12-layer models (GPT, LLAMA, BERT, T5) to 20k steps. We included the updated results in Section 4.2 and Figure 3. Additionally, it's worth noting that the theoretical correctness of $\mu$P [1] is independent of the number of training steps.
> ### Q3: The paper lacks its own insights into the scaling law compared with main components.
> We believe there are misunderstandings:
> (1) We wish to clarify that nanoLM is an extension of $\mu$scaling [1] (preprint on arXiv). We've contacted the area chair for more details and will update upon receiving feedback. Specifically, Unlike $\mu$Scaling uses 8M-676M GPT models to predict the loss for 1.4B models, nanoLM extends this to 77M-3.4B for predicting loss in 52B models. This scale is 184x larger than $\mu$Scaling, marking a critical step for LLMs. And nanoLM providing extensive datasets ranging from 100B to 2T for comprehensive testing. Additionally, nanoLM supports a variety of architectures, including decoder-only structures (e.g., GPT, LLAMA), encoder-only structures (e.g., BERT), and encoder-decoder structures (e.g., T5), and is compatible with data parallelism. These enhancements are particularly crucial for researchers in the field of LLMs, offering a more robust benchmark for researcher.
> (2) The key distinction of nanoLM from prior work is its **avoidance of re-searched on hyperparameter (HP) for large language models (LLMs).**  There are very few related work to ours. Except those mentioned in the paper, we (as well as [4]) noticed a contemporary work [3] that applies $\mu$P in language model pre-training. Their main finding is that $\mu$P results in a constant (thus predictable) deviation from the scaling laws fitted with Standard Parametrization (SP). However, SP scaling laws need HP search on large scales because they have large variances. Besides, they varied some non-$\mu$Transferable HPs for different model scales, while we only vary the widths and show that the $\mu$P process itself results in a model sequence with accurate loss prediction. For these two reasons, the stories are very different.
> Thank you for pointing out that this comparative analysis is unclear; we updated the above-mentioned analysis in new Appendix A to clarify this.
> ### Q4: The paper need phrase the critical component of the method more clearly.
> We agree. We clarified the key elements of our method in Section 3.
> ### Q5: Loss prediction values and ground truth should put into a table in the main paper.
> Thank you for the suggestion. We incorporated loss prediction values and ground truth in new Table 3 of the main paper.
> ### Q6: Typos.
> We're sorry for these mistakes. We have rechecked the paper and corrected the typos.
> ### ref
> [1] https://arxiv.org/abs/2011.14522
> [2] https://arxiv.org/pdf/2304.06875.pdf
> [3]https://arxiv.org/pdf/2304.03208.pdf
> [4] https://www.semianalysis.com/p/google-we-have-no-moat-and-neither

---

> > ### Comment · Reviewer_Fxnv · 2023-11-21
> >
> > I have read the author's responses and would like to keep my original score.

---

> > > ### Author Response · Authors · 2023-11-23
> > > **Thank You for Reviewing nanoLM.**
> > >
> > > Dear Fxnv,
> > >
> > > Thank you for reviewing our nanoLM  paper. Your comments and insights have been greatly appreciated.
> > >
> > > We respect and value your assessment and are grateful for the time you spent on our work.
> > >
> > > Best regards, The authors

---

### Official Review · Reviewer_z2mu · 2023-10-30

**Soundness:** 2 fair
**Presentation:** 3 good
**Contribution:** 2 fair
**Rating:** 5
**Confidence:** 3

**Summary:**

The study addresses the challenge of identifying the best model configurations when computational resources are limited. It applies a method that involves training smaller models to estimate the performance of larger ones, thus facilitating the selection of the most efficient model design.

**Strengths:**

1. The paper is well-constructed, with lucidity and straightforwardness. The rationale behind selecting the most effective model design under computational constraints is well-motivated.
2. The authors have successfully utilized established techniques such as \mu P and \mu Scaling to anticipate the performance loss in larger Language Models (LLMs), thereby economizing on training expenses.
3. The research contributes to the field by making available a comprehensive pre-training dataset encompassing 100B to 2T tokens.

**Weaknesses:**

1. The paper's novelty is questioned as the methodology seems to be a synthesis of pre-existing approaches.
2. The datasets introduced appear to be aggregates of data already available.
3. The study does not offer an in-depth comparative analysis or discussion on how their proposed methodologies diverge from OpenAI's established scaling law[1,2] for loss prediction, as detailed in prior technical reports.

[1] GPT-4 Technical Report

[2] Henighan, Tom, et al. "Scaling laws for autoregressive generative modeling." arXiv preprint arXiv:2010.14701 (2020).

**Questions:**

See weaknesses.

---

> ### Author Response · Authors · 2023-11-17
> **Thank you for your valuable suggestions.**
>
> Thank you kindly for your valuable suggestions. Following your advice, we offer clarifications to better explain our paper's novelty and contributions. We address each of your points in detail below.
> ### Q1: The paper's novelty is questioned as it merges existing methods.
> Thank you for your thoughtful review and for raising concerns about the novelty of our work.  We wish to clarify that nanoLM is an extension of $\mu$scaling [1] (preprint on arXiv). We've contacted the area chair for more details and will update upon receiving feedback. Unlike $\mu$Scaling uses 8M-676M GPT models to predict the loss for 1.4B models, nanoLM extends this to 77M-3.4B for predicting loss in 52B models. This scale is **184x** larger than $\mu$Scaling, marking a critical step for LLMs. Additionally, nanoLM supports various architectures, such as decoder-only (e.g., GPT, Llama), encoder-only (e.g., BERT), and encoder-decoder (e.g., T5), and is data parallelism compatible. These features are vital for LLM researchers, offering a more comprehensive benchmark.
> ### Q2: The datasets introduced appear to be aggregates of data already available.
> Thank you for pointing this out. The primary intention behind curating and categorizing datasets in the nanoLM benchmark is to empower researchers to conduct meaningful comparisons between different model architectures and algorithms, especially when operating under computational constraints. We acknowledge that the provision of datasets is not an innovation in itself, but rather a facilitation for researchers.
> ### Q3: Comparative analysis diverge from OpenAI's established scaling law for loss prediction.
> Thank you for pointing out the inadequacies in this part of our analysis. We add comparisons and experments with OpenAI's scaling law from the following perspectives:
> (1) GPT-4 [3] technical report shows that some behaviors of large models can be accurately predicted before the training starts. However, GPT-4 has not disclosed the techniques used for its loss prediction, so we are unable to make a comparison with it.
> (2) General conditions for scaling laws? Previous scaling laws [4] directly search for HPs on each scale, and the optimal HPs do not satisfy $\mu$P[2] function. This indicates that $\mu$P is a sufficient but not necessary condition for scaling laws, and scaling law itself may represent a higher level of universality.
> (3) The key distinction of nanoLM from prior work is its **avoidance of repeated hyperparameter (HP) searches for large language models (LLMs)**. There are only a few works similar to ours. Beyond those cited in our paper, we, along with [6], recognize a concurrent study [5] applying $\mu$P in language model pre-training. Their main discovery is that $\mu$P leads to a predictable deviation from scaling laws established with Standard Parametrization (SP). However, SP scaling laws demand HP searches at larger scales due to high variances. Additionally, they modified some non-$\mu$Transferable HPs across different model sizes. In contrast, we vary only the widths and demonstrate that $\mu$P alone can produce a model sequence with accurate loss prediction.  For these two reasons, the stories are very different.
> (4) Embedding counts as model size. We add experiment shows that our scaling law fits less accurately if embedding weights are excluded, unlike OpenAI established scaling law [4]. This could be due to $\mu$P's finding that embedding layer learning rates shouldn't be scaled down with widths. While [4] unified learning rate for all layers at each model size might result in slower learning for embeddings, with matrix-like parameters dominate the training dynamics.
> Thank you for pointing out that this comparative analysis is unclear; we will update the above-mentioned experiment and analysis in new Appendix to clarify this.
> ### ref
> [1] https://arxiv.org/pdf/2304.06875.pdf
> [2] https://arxiv.org/abs/2011.14522
> [3] https://arxiv.org/pdf/2303.08774.pdf
> [4] https://arxiv.org/abs/2010.14701
> [5] https://arxiv.org/pdf/2304.03208.pdf
> [6] https://www.semianalysis.com/p/google-we-have-no-moat-and-neither

---

> > ### Comment · Reviewer_z2mu · 2023-11-20
> >
> > Thanks for the reply. After thoroughly reviewing the provided rebuttal and considering other reviews, I have decided to maintain my original score.

---

> ### Author Response · Authors · 2023-11-22
> **Thank you for your prompt response!**
>
> We really appreciate you getting back to us so soon. We would like to clarify a previous point, as there seems to have been some misunderstanding. We wish to emphasize that nanoLM is indeed the **latest version of our $\mu$scaling work (preprint on arXiv)**, although it has not yet been updated on arXiv. We believe this is crucial in highlighting the novelty of our work. Thank you again for your attention and valuable feedback on our work.

---

### Official Review · Reviewer_eip6 · 2023-10-31

**Soundness:** 3 good
**Presentation:** 3 good
**Contribution:** 3 good
**Rating:** 6
**Confidence:** 3

**Summary:**

Large language models (LLMs) have shown impressive performance on various tasks, with models getting increasingly larger. However, training such large models is computationally expensive. Recent work has explored scaling up data size rather than just model size, showing performance gains with smaller models trained on more data (Chinchilla). The goal of this paper is to introduce nanoLM, a benchmark for cost-effective LLM research within limited compute budgets. Larger LLMs like Meta's LLAMA require upwards of 1.7 million GPU hours and hundreds of billions of tokens to pre-train, using model parallelism.
Authors rely on μP and μScaling, which are methods for transferring hyperparameters (HPs) when scaling up model size. μP is a zero-shot transferring function for certain HPs like learning rate and initialization variance when changing model width. μScaling complements μP by fitting a power-law function to predict the loss L' that could be achieved by training a larger model M' with width w' and HP H', without directly training M'.
Together, μP and μScaling allow extrapolating language models from small to large scale without expensive direct training, by predicting the optimal HPs and loss for larger models based on results from smaller models. This enables more efficient study of how model structure and non-μTransferable parameters affect large-scale LMs.
Authors then introduce the NanoLM benchmark for computing the loss of the different models with varying sizes. NanoLM allows comparing different LLM architectures by predicting their training loss, without having to do full training. It supports common architectures like GPT, BERT, and T5. Experiments validate nanoLM's ability to accurately predict loss for large models (26B-52B parameters) by fitting small models (38M-3.4B parameters), using just 13-14% of the total pre-training cost. They also include other empirical evaluations of the released benchmark. Finally, the authors release curated pre-training datasets with 100B to 2T tokens covering various domains. All code and datasets are open-sourced.

Overall this is a potentially high impact paper since releasing benchmarking datasets is always a great addition to the field. However, I think there is some analysis missing on comparison of their work with other similar papers such as  [https://arxiv.org/pdf/2304.01373.pdf, https://arxiv.org/pdf/2309.14322.pdf]. I would like to see more evidence that the predictions errors are meaningfully small and therefore the results are robust and would be happy to update my review if those are provided,

**Strengths:**

- Authors tackle an important problem of democratizing access to effectively experimenting with and comparing large language models, which has so far been limited to only the most resource-rich organizations. This could help advance LLM research and applications. It Enables researchers to compare LLMs within limited compute by predicting losses of smaller models instead of expensive end-to-end training.
- Authors open source their benchmark (and code) thus greatly contributing to the advancement of the field. Making more benchmarks available ensures the field continues to innovate on interesting problems and ensures the available datasets are not overused.
- Authors validates nanoLM's loss prediction capabilities across scales - from simplified settings to 26B and 52B parameter models and show it achieves accurate loss forecasting for large models by fitting losses of smaller models, reducing pre training costs significantly (to 13-14% of total).
- Authors release the curated datasets used for pre-training and evaluation which can be used by other researchers.

**Weaknesses:**

- The paper relies heavily on loss prediction as an evaluation metric, but does not provide strong evidence that lower loss directly translates to better downstream task performance. More analysis is needed to validate that loss is an appropriate proxy.
- Authors show experiments that indicate that nanoLM can predict the loss of extremely large-sized models by fitting the losses of several smaller models. It is not clear how to calibrate the error they report. It would be good to compare the results the authors show the comparison with other sources of data for predicting loss such as the results of [https://arxiv.org/pdf/2304.01373.pdf]. In the same vein, it would be good to compare against other work in the field e.g. [https://arxiv.org/pdf/2309.14322.pdf].
- The long-term value of the benchmark requires ongoing maintenance and user adoption, which are not discussed. Plans for supporting and expanding nanoLM could be elaborated.

**Questions:**

- The experiments are limited to English language modeling. Have the authors considered testing the approach for other languages and tasks? It would strengthen the claims of applicability.
- How can I calibrate the results of this work, could you elaborate on why the reported errors are considered small? I think there is some analysis missing on comparison of their work with other similar papers such as  [https://arxiv.org/pdf/2304.01373.pdf, https://arxiv.org/pdf/2309.14322.pdf]. I would like to see more evidence that the predictions errors are meaningfully small and therefore the results are robust.

---

> ### Author Response · Authors · 2023-11-17
> **Thank you for your constructive feedbacks**
>
> We have added additional experiments for the important questions asked in the review. We believe these experiments have strengthened the paper’s robustness.
> ### Q1: Experiments are limited to English language modeling. Plans for suporting/expanding nanoLM are not discussed.
> Thank you for your suggestions. $\mu$P's [2] thereactical proof shows nanoLM suits most transformer-like architectures. We've included Llama[1] and other language in nanoLM's benchmark for loss prediction, demonstrating its robustness; These updates are detailed in the revised Section 4.2 and Figure 3. In the future, we plan to integrate additional popular TF-like architectures and tasks (eg., vision) into the nanoLM and encourage users to test, aiding in the continuous evolution of the nanoLM benchmark.
> ### Q2: More evidence to explain why the predictions errors are meaningfully small.
> Thank you for raising this important question and for providing related papers [3][4] that aid in our explanation. Paper [3] primarily explores the relationship between learning rate and training stability, without focusing on loss prediction, a key aspect of our work. In contrast, nanoLM conducts the $\mu$HPs search on smaller models to ensure entry into the loss basin, facilitating accurate predictions. Additionally, the largest model in [3] is only 1.2B, making direct comparisons with our larger models impractical. We have conducted additional experiments to further calibrate the prediction errors of nanoLM, as detailed below:
> (1) Based on Table 8 of the Cerebras-GPT [5] and the scaling law's power function $L=ax^b+c$, we conducted an error comparison analysis of the predicted losses for Cerebras-GPT[5] , Pythia[4], and nanoLM. Cerebras-GPT fitted the 13B model's loss using 111M-6.7B models, Pythia for 12B with 7M-6.9B, and nanoLM for 52B from 77M-3.4B, with their respective errors being **0.025, 0.019, and 0.022**. When fitting losses for models down to 10B, the errors are **0.034, 0.049, and 0.018**, respectively. Additionally, we calculated the covariances of the fitted coefficients {a, b, c}, finding that **nanoLM's covariances are significantly lower** than those of Cerebras-GPT and Pythia. These experimental results  demonstrate the (a) $\mu$P infinite neural network is theoretically correct and (b) scaling laws are empirically reliable on any scale. (c) The loss prediction of nanoLM is more stable and reliable. Loss prediction validation is costly, and we have conducted as many experiments as possible within our computational power limits (nanoLM has reached 52B for this purpose, while Pythia and Cerebras-GPT have only reached 13B). Further experimentation would be prohibitively expensive in terms of computational costs.
> (2) **Small models are more vulnerable and can be solved by average fitting**. We predict loss of small model (6-layer models). After grid search for the best $\mu$HPs being inside loss basin. nanoLM works perfectly for this $\mu$HPs. We then explored other $\mu$HPs around it and found that the scaling laws have larger deviations than 12-layer models. This is potentially because small models are more vulnerable to slight misalignment of loss landscapes across $\mu$-Transfer. However, we can easily balance-off this deviation by fitting scaling laws with the average results across all these HPs near the loss basin. This can be very practical because we observe that larger widths have low variance in training loss w.r.t different $\mu$HPs.
> We presented these analyses in the updated Appendix A.
> ### Q3: Is pre-training loss an appropriate proxy for assessing LLM performance effectiveness?
> Great question! The effectiveness of LLMs stems from their base models, with the absolute value of loss being pivotal in fixed datasets. This is evidenced by GPT-4 [6], which targets a loss value of 1.25. Additionally, LLMs' training involves language modeling tasks, like predicting subsequent or masked tokens. This loss strongly correlates with perplexity. LLaMA [1] also suggests a link between pre-training loss and downstream performance, although this is challenging to assess comprehensively. As shown in LLAMA's Figure 1, there's a notable decline in training loss across 7B-65B models. Figure 2 further demonstrates enhanced performance in question answering and common sense reasoning with increased training tokens. Thus, pre-training loss generally aligns closely with downstream performance, underscoring the significance of nanoLM's loss prediction.
> Thank you for pointing out that this analysis is unclear; we added the above-mentioned analysis in Section 3.1 to clarify further.
> ### ref
> [1]https://arxiv.org/pdf/2302.13971.pdf
> [2]https://arxiv.org/abs/2011.14522
> [3]https://arxiv.org/pdf/2309.14322.pdf%5D
> [4]https://arxiv.org/pdf/2304.01373.pdf
> [5]https://arxiv.org/pdf/2304.03208.pdf
> [6]https://arxiv.org/pdf/2303.08774.pdf

---

> ### Comment · Reviewer_eip6 · 2023-11-22
>
> Dear Authors,
>
> Thank you for the extensive response, I think my questions 1 and 2 are addressed. However regarding question 2 I am still somewhat conflicted - what I want to understand is how difficult of a task is error rate predictions? I suppose I would like to see a discussion about what other baselines we could use (perhaps simpler formulas, random guess, etc.) to calibrate what is a small error score and what would be a larger error score. I understand that this request comes very late but I think it would be still a valuable addition to the paper.
>
> In light of answers 1+3 I am happy to imporve my score to 6.
>
> Best,
> reviewer eip6

---

> ### Author Response · Authors · 2023-11-23
> **Thank You for the Revised Review on nanoLM!**
>
> Dear reviewer eip6,
>
> We are grateful for your updated review and the improved rating for our paper. Your feedback is very valuable to us.
> We will quickly carry out the additional baseline experiments you suggested and include them in our work.
>
> Thank you again for your helpful comments and support.
>
> Best regards,
> The authors

---

### Author Response · Authors · 2023-11-20
**Rebuttal Summary**

We thank all the reviewers for their detailed feedback and positive recommendation for our work. We are glad that **eip6** found our work "as a potentially high impact paper."
We tried to answer all the requests from the reviewers. We run the suggested experiments – as a result we have **a better understanding of the nanoLM robustness and insight to scaling law**. Important highlights from this rebuttal period were:
* We have enhanced the nanoLM benchmark by adding **LLaMA and other languages**, extending training steps to bolster its robustness, and outlined future plans for supporting and developing nanoLM further. (Figure 3, Section 4.2, Conclusion and future work)
* A detailed **error comparison** of nanoLM is presented, setting it against similar work like Pythia and Cerebras-GPT. (Appendix A)
* We explain nanoLM's unique perspectives on scaling laws, including scaling law fail outside the loss basins; smaller models are more vulnerable; embedding counts in model size. (Appendix A)
* Highlighting nanoLM's novelty, particularly in contrast to $\mu$P and $\mu$Scaling. (Section 2.2)
* We restructured the methodology, enhancing crucial components and providing explanations for the relationship between pre-training loss and model performance. (Section 3.1)

---

### Meta-Review · Area_Chair_CPFr · 2023-12-03

**Metareview:**

The paper investigates the question on how the performance of an language model can be predicted with computational constraints. For this, the paper proposes a framework called NanoLM.

The paper received four reviews (5,5,6,6). Both reviewers z2mu and Fxnv remark that the contribution of the paper relative to prior work is unclear. In response, the authors note that this work is an extension of their prior work [1] (​​https://arxiv.org/pdf/2304.06875.pdf) thus revealing authorship. It is inappropriate to reveal authorship during the double-blind review process.

Based on the reviews and my carful reading, in my opinion, the concern is not novelty relative to the paper [2] but relative to existing scaling law papers and works that enable scaling while keeping the parameters constant (like mu-parameterization, and the tensor-program papers). Those papers already enable computationally efficient loss prediction to some extend, and loss-prediction with training smaller models and using scaling laws to predict the performance of larger ones is somewhat standard in the field. While loss prediction under computational constraints remains an issue, it is unclear to me how this paper enables loss-prediction beyond what we can predict already through existing scaling laws and through scaling networks with mu-parameterization (see Tensor Programs V paper) and extensions of mu-parameterization that enable hyperparameter transfer when scaling.

**Justification For Why Not Higher Score:**

The paper is not sound and the contribution relative to prior work is unclear.

**Justification For Why Not Lower Score:**

N/A

---

### Decision · Program_Chairs · 2024-01-16

Reject